# Milk Intake in Early Life and Later Cancer Risk: A Meta-Analysis

**DOI:** 10.3390/nu14061233

**Published:** 2022-03-15

**Authors:** Hyeonmin Gil, Qiao-Yi Chen, Jaewon Khil, Jihyun Park, Gyumi Na, Donghoon Lee, Nana Keum

**Affiliations:** 1Department of Food Science and Biotechnology, Dongguk University, Goyang 10326, Korea; gusals99@dgu.ac.kr (H.G.); chenqiaoyi0505@naver.com (Q.-Y.C.); kyk3079@naver.com (J.K.); jyun4071@naver.com (J.P.); gyumi177@gmail.com (G.N.); 2Department of Nutrition, Harvard T.H. Chan School of Public Health, Building 2, 3rd Floor, 665 Huntington Avenue, Boston, MA 02138, USA; dhlee@mail.harvard.edu

**Keywords:** childhood, adolescence, milk intake, breast cancer, prostate cancer, colorectal cancer, meta-analysis

## Abstract

Dairy consumption in adulthood has been demonstrated to influence cancer risk. Although childhood and adolescence represent critical periods of rapid growth, the relationship between milk intake in early life and later cancer risk is unclear. Thus, we examined this relationship by conducting a meta-analysis of the observational studies. PubMed and Embase were searched for relevant articles that were published throughout December 2021. The summary relative risk (RR) and 95% confidence interval (CI) were estimated using the DerSimonian-Laird random-effects model. The summary RR for the highest vs. lowest milk intake was 0.83 (95% CI = 0.69–1.00; *p* = 0.05; I^2^ = 60%; seven studies) for breast cancer, 0.98 (95% CI = 0.72–1.32; *p* = 0.88; I^2^ = 51%; four studies) for prostate cancer, and 0.90 (95% CI = 0.42–1.93; *p* = 0.78; I^2^ = 83%; three studies) for colorectal cancer. No evidence of an association emerged in subgroup analyses of menopausal status, cancer stage, fat content of milk, life stage of milk intake, or study design. Consistent results were observed in the meta-analyses using total dairy intake. In conclusion, milk intake during childhood and adolescence might not be associated with risks of breast, prostate, and colorectal cancer later in life. Given the small number of studies that were included in our meta-analysis, and the high heterogeneity, more studies are warranted for a definitive conclusion.

## 1. Introduction

Dairy products are widely consumed worldwide, and they contain diverse components that are shown to influence cancer risk. The most notable one is calcium, as a high calcium intake is shown to suppress prostate cancer proliferation [1,2], while promoting colorectal epithelial cell differentiation and apoptosis [3]. Dairy products also contain insulin-like growth factor-I (IGF-I) [4], and for adults, a high dairy intake is associated with an increase in IGF-I concentration [5], stimulating cell proliferation. Indeed, a recent meta-analysis has shown that high milk intakes in adulthood were associated with an increased prostate cancer risk, but a decreased colorectal cancer risk [6]. However, dairy intakes were not associated with a risk of other cancers including breast cancer [7], pancreatic cancer [8], lung cancer [8], and ovarian cancer [8]. Thus, although dairy consumption is differentially associated with cancer risk by cancer type, it is nevertheless implicated in the development of some cancers in adults.

Milk and other dairy products are also widely consumed in early life, with dietary guidelines for Americans recommending that children and adolescents consume the equivalent of 2–3 cups of dairy products per day depending on their age [9]. Of note, the development and progression of cancer spans several decades, and childhood and adolescence are marked by rapid cell division and growth [10]. Thus, food intake during these life stages may have a considerable influence on the early stages of carcinogenesis. This premise is corroborated by observations that established risk factors of cancers, such as tallness and early age at menarche, are affected by childhood and adolescent diet [11]. In fact, calcium and IGF-1 in dairy products play critical roles in development and growth during childhood and adolescence [12], and several studies have investigated how dairy intake in early life relates to cancer risk [2,11,13,14,15,16,17,18,19,20,21,22,23,24,25,26,27]. However, the results are inconsistent, and no meta-analysis has been conducted yet. Thus, we performed a meta-analysis to systematically summarize the relationships between cancer risk and the intake of milk and other dairy products.

## 2. Materials and Methods

The design, analysis and reporting of this meta-analysis was performed in accordance with the Meta-analysis Of Observational Studies in Epidemiology (MOOSE) (Appendix A) [28].

### 2.1. Study Search

PubMed and Embase were searched for relevant articles published throughout December 2021. The search was limited to human studies and the English language, but no other restrictions were imposed. The detailed search terms are provided in Appendix A.

### 2.2. Study Selection

To be included, studies had to be an observational study (e.g., case-control study, cohort study) that examined the relationship between adolescent dairy intake (i.e., exposure) and cancer risk later in life (i.e., outcome), with the results reported as relative risks (RR) (odds ratio, rate ratio, hazard ratio) and at a 95% confidence interval (CI). We excluded abstracts, unpublished results, and review articles. When there were multiple publications from the same cohort [11,21,22,29], we chose the most recent article [21,22]. To identify additional papers, the reference lists of selected articles and previous systematic reviews were reviewed. After a series of screening, a total of 15 studies (9 studies for breast cancer, 3 studies for prostate cancer, 2 studies for colorectal cancer, 1 study for breast, prostate, and colorectal cancer) were eligible for this meta-analysis [2,13,14,15,16,18,19,20,21,22,23,24,25,26,27]. This study selection process is summarized in Figure 1. The characteristic of studies included are presented in Appendix A.

### 2.3. Data Abstraction

The following information was extracted from each article: name of the first author, publication year, study design, type (milk, dairy), fat content (low-fat, regular), and life stage (in age) of dairy intake; type and subtype (e.g., by stage) of cancer, relative risk and corresponding 95% CI, important characteristics of the study population (e.g., country, age, sex), and variables which were adjusted for.

For the study selection and data extraction, two authors (HG and Q-YC) independently participated, and any disagreement was resolved through discussion with NK.

### 2.4. Statistical Analyses

Across studies, milk was the principal type of dairy intake, and the total dairy intake was defined heterogeneously. Thus, we set milk as the primary exposure and ran sensitivity analyses with dairy intake. The summary RR and the 95% CI were calculated using the DerSimonian-Laird random effects model, which estimates the weighted mean of true effects, and accounts for both between-study variations and within-study variations [30]. Potential for small study effects, such as publication bias (i.e., bias arising from the selective publication of studies, with small studies that produced nonsignificant results being less likely to be published), was assessed using Egger’s test [31]. In the test that regresses the standardized effect sizes on their precisions, the null hypothesis of no publication bias is equivalent to the regression intercept of zero [32]. When evidence for small study effects was indicated, we plotted a funnel plot that incorporates contours of statistical significance [33]. The contour-enhanced plot enables us to check the statistical significance of missing studies, as imputed by the trim-and-fill method, thus aiding the understanding of sources relating to funnel plot asymmetry [33].

Heterogeneity in the relationship across trials was assessed by I^2^ [34], which represents the percentage of total variation across studies that is attributable to between-study heterogeneity [35]. An I^2^ value of 25%, 50%, and 75% was considered to represent low, moderate, and high heterogeneity, respectively [35]. Potential sources of heterogeneity were explored based on a priori selected variables related to (1) etiologic heterogeneity (menopausal status: premenopausal vs. postmenopausal for breast cancer, cancer stage: invasive vs. any for breast cancer, advanced vs. any for prostate cancer), (2) potential effect modifiers (fat content: low-fat milk vs. regular milk, life stage of milk intake, Age (years): <10–12 vs. 10–12 ≤ Age (years) ≤ 19), and (3) methodological characteristics (study design: retrospective study vs. prospective study). Considering that summary results from a meta-analysis of a small number of studies are not informative, heterogeneity was explored when the total number of studies were greater than three.

All the statistical tests were two-sided and *p* values of < 0.05 were considered statistically significant. Analyses were performed using STATA 17 (StataCorp, College Station, TX, USA).

## 3. Results

### 3.1. Milk Intake and Breast Cancer Risk

A total of seven studies were included in this meta-analysis [2,19,22,23,25,26,27]. The summary RR of breast cancer for highest vs. lowest milk intake in childhood and adolescence was 0.83 (95% CI = 0.69–1.00, *p* = 0.050), with moderate heterogeneity (I^2^ = 60%) (Figure 2A). Small study effects, such as publication bias, were indicated by funnel plot asymmetry (Appendix A) and Egger’s test (P_Egger_ = 0.04). Several sensitivity analyses were performed. First, when meta-analysis was run after omitting one study that showed the most extreme result (i.e., the study that lay outside of the pseudo 95% confidence limits) [26], the summary RR was 0.91 (95% CI = 0.79–1.05, I^2^ = 31%) with no evidence of small study effects (P_Egger_ = 0.06). Second, the use of the contour-enhanced funnel plot (Appendix A) potentially missed studies imputed by the trim-and-fill method as they were located in both statistically significant and non-significant areas, which suggests that the observed asymmetry is not solely caused by publication bias based on statistical significance.

When heterogeneity was explored, there was no evidence of a significant relationship between early life milk intake and breast cancer risk, regardless of menopausal status (Figure 2B), cancer stage (Figure 2C), fat content of milk (Figure 2D), life stage of milk intake (Figure 2E), and study design (Figure 2F).

When sensitivity analyses were conducted upon dairy intake in early life rather than milk intake, the aforementioned results remained consistent (Appendix A).

### 3.2. Milk Intake and Prostate Cancer Risk

A total of four studies were included in this meta-analysis [2,13,14,15]. The summary RR of prostate cancer for the highest vs. lowest milk intake in childhood and adolescence was 0.98 (95% CI = 0.72–1.32), with moderate heterogeneity (I^2^ = 51%) (Figure 3A). There was no evidence of small study effects such as publication bias (P_Egger_ = 0.52).

When heterogeneity was explored, there was no evidence of a significant relationship between early life milk intake and prostate cancer risk, with respect to cancer stage (Figure 3B), life stage of milk intake (Figure 3C), and study design (Figure 3D). Heterogeneity by fat content of milk was not performed, because no study examined the effect of low-fat milk intake in early life on prostate cancer risk in adulthood.

In sensitivity analyses based on early life dairy intake, the aforementioned results remained consistent (Appendix A).

### 3.3. Milk Intake and Colorectal Cancer Risk

A total of three studies were included in this meta-analysis [2,16,18]. The summary RR of colorectal cancer for highest vs. lowest milk intake in childhood and adolescence was 0.90 (95% CI = 0.42–1.93), with high heterogeneity (I^2^ = 83%) (Figure 4). Small study effects, such as publication bias, were not indicated by Egger’s test (P_Egger_ = 0.93). In sensitivity analysis based on early life dairy intake, no significant association was observed (summary RR = 1.07, 95% CI = 0.59–1.94, I^2^ = 77%) (Appendix A).

Despite the high heterogeneity observed, due to a small number of studies, heterogeneity of the a priori selected factors was not explored.

## 4. Discussion

In this meta-analysis of observational studies, we observed that childhood and adolescent dairy intake, as represented by milk intake, was not associated with the risks of breast, prostate, and colorectal cancers in later life. Furthermore, the relationships did not vary by menopausal status, cancer stage, fat content of milk, life stage of milk intake, and study design. Although there was suggestive evidence of a reduced breast cancer risk associated with a higher milk intake in early life, the association was not statistically significant and was affected by publication bias. Thus, our study suggests that dairy intake during childhood and adolescence might neither benefit nor harm one’s cancer risk during adulthood.

With regard to breast cancer risk, out of the seven studies included in our meta-analysis, six studies [2,19,22,23,25,26] were conducted in western countries, most of which found no evidence of a significant relationship. In contrast, the remaining study [27] was conducted in China and it observed a significantly reduced risk of breast cancer in relation to a higher milk intake in early life. This significant inverse association might be partly attributed to selection bias or recall bias in case-control studies. However, we cannot rule out the possibility that true heterogeneity in the relationship exists across different racial groups with different genetic make-ups and dietary patterns. For instance, milk intake leads to an increase in IGF-I level [5], which in turn may contribute to breast cancer development and progression [36]. Indeed, in a cross-sectional study conducted in the U.S., an association between milk intake and circulating IGF-I levels differed by race and ethnicity [37]. Furthermore, regarding dairy intake in adulthood, a meta-analysis observed that a significant inverse association with breast cancer risk was pronounced in Asian populations compared with Western populations [38].

It has been suggested that several compounds in milk and other dairy products confer protection against breast cancer. Dairy products are the main food source of calcium in many populations. Intracellular calcium regulates the synthesis of components related to cell proliferation and differentiation [39,40,41], and high calcium intake was shown to suppress hyperproliferation of epithelial mammary cells in mice [42]. Dairy products are often fortified with vitamin D, which plays an anticarcinogenic role by binding to a vitamin D receptor in breast epithelial cells, which in turn promotes the transcription of genes that are involved in having antiproliferative, prodifferentiating, and proapoptotic effects on cells [43]. Of note, in nutritional supplementation, individuals with insufficient micronutrient intakes are expected to benefit more from an additional intake, compared with individuals who have sufficient micronutrient intakes [44]. Given that the average calcium intake is generally lower in Asian populations than Western populations, additional calcium intake resulting from higher milk intake might elicit a pronounced benefit against breast cancer risk in Asian populations.

For colorectal cancer, given that the evidence for dairy products and calcium supplements protecting against colorectal cancer in adults is considered to be strong by the Continuous Update Project (CUP) panel [45], calcium has been suggested as a main component that mediates the beneficial effects of dairy intake on colorectal cancer risk [3]. Calcium in the intestinal lumen binds and inactivates carcinogenic materials (e.g., secondary bile acids, ionized fatty acids), and intracellular calcium promotes colorectal epithelial cell differentiation and apoptosis, all of which contributes to a reduction in colorectal cancer risk [3]. However, dairy products contain IGF-1, and a high milk intake is associated with an increase in circulating IGF-1 levels in children as well as in adults [46]. IGF-1 promotes colorectal carcinogenesis by stimulating cell proliferation and inhibiting apoptosis [3]. In light of the fact that childhood and adolescence are marked by rapid growth, consistent exposure to high levels of circulating IGF-1 may set the stage for cancer development and progression in later life. Taken together, because dairy products contain both anti-carcinogenic and carcinogenic compounds, their effects on colorectal cancer risk are multifaceted, and this may explain the high heterogeneity observed in our meta-analysis of milk intake and colorectal cancer risk, with two studies finding significant protective effects [16,18], whereas one study observed a significant adverse effect [2].

Concerning prostate cancer, it is notable that calcium, unlike its anti-carcinogenic effect on breast and colorectal cancers, may rather increase prostate cancer risk by inhibiting the synthesis of 1,25-dihydroxyvitamin D3 (1,25(OH)2D3, also known as calcitriol), and the active metabolite of vitamin D3 that suppresses cell proliferation in the prostate [1]. Adolescent milk intake increases circulating IGF-1 levels during the critical period of prostate development and growth, thereby promoting prostate cell proliferation and enhancing the survival of partially transformed cells which would otherwise be forced into apoptosis [47]. Saturated fat, another main component of milk, creates reactive oxygen species that damage the tumor-suppressing activity of the p53 protein, thereby inducing prostate cancer [48]. Finally, cow milk contains high levels of microRNAs [49], which are transferred to human cells and suppress the expression of TP53, which is the gene that produces the p53 protein [50]. Indeed, a recent meta-analysis found that high intakes of dairy products, and dairy calcium in adults, were associated with an increased prostate cancer risk [51]. Despite the fact that multiple components in dairy products promote prostate carcinogenesis, high dairy intake in early life was not associated with later prostate cancer risk in our study. More studies are warranted to elucidate the mechanism, but a potential explanation includes an inverse association between childhood dairy intake and adult IGF-1 concentration, which might mitigate the aforementioned carcinogenic effects of dairy intake.

There are several strengths to our meta-analysis. To our knowledge, this is the first meta-analysis on the relationship between milk intake during childhood and adolescence, and cancer risk later in life. By conducting subgroup analyses by menopause, cancer stage, or life stage of milk intake, when possible, our study explored the potential for important heterogeneity in the relationship.

However, our study has limitations to acknowledge. First, the number of studies included in our meta-analysis was small, particularly for colorectal cancer (*n* = 3). Thus, despite the high heterogeneity observed for colorectal cancer, we could not explore the sources of heterogeneity. Second, the summary estimate for breast cancer was affected by small study effects, which reduces the validity of our finding. However, we conducted several sensitivity analyses to check the consistency of the results. Third, as a meta-analysis, our results are also affected by biases that individual studies included in our meta-analysis are prone to, including confounding. Finally, we could not address important questions such as the interaction between milk intake in early life, the intake in adulthood, and heterogeneity in breast cancer by hormone receptor status, due to lack of data from individual studies included in our meta-analysis.

## 5. Conclusions

In conclusion, unlike well-known associations between adult dairy intake and cancer risk, intake of milk and other dairy products during childhood and adolescence might not be associated with a cancer risk for the breast, prostate, and colorectum later in life. However, due to a limited number of studies included in our meta-analyses, our results need to be interpreted with caution until they are confirmed by future studies. Future studies are warranted to address the potential interaction between dairy intake in early life and that in adulthood in relation to cancer risk. 

## Figures and Tables

**Figure 1 nutrients-14-01233-f001:**
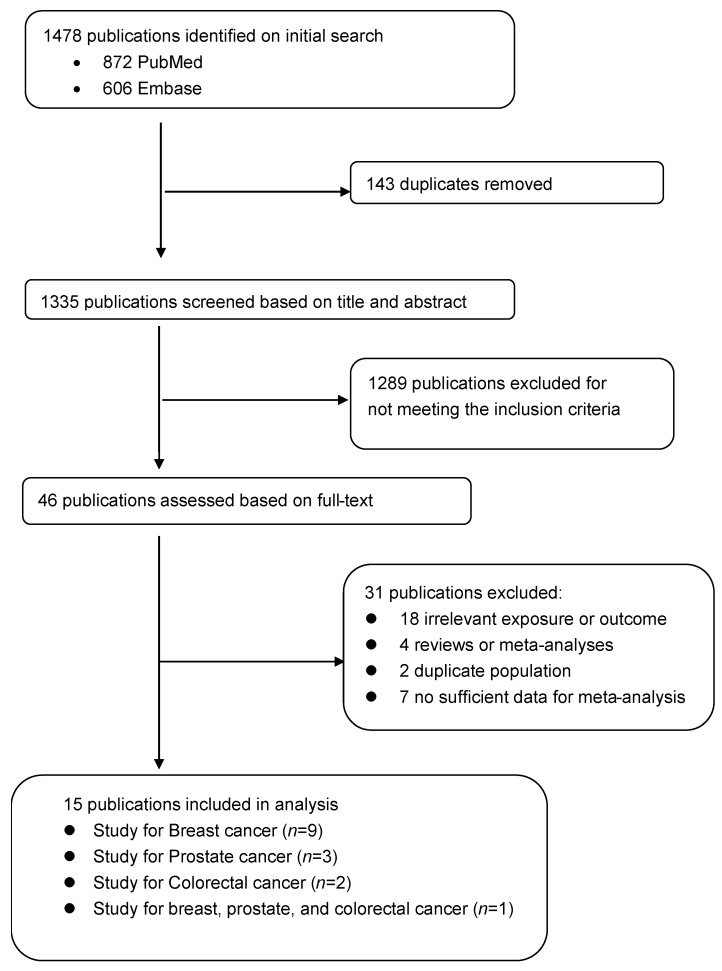
Flowchart for Study Selection.

**Figure 2 nutrients-14-01233-f002:**
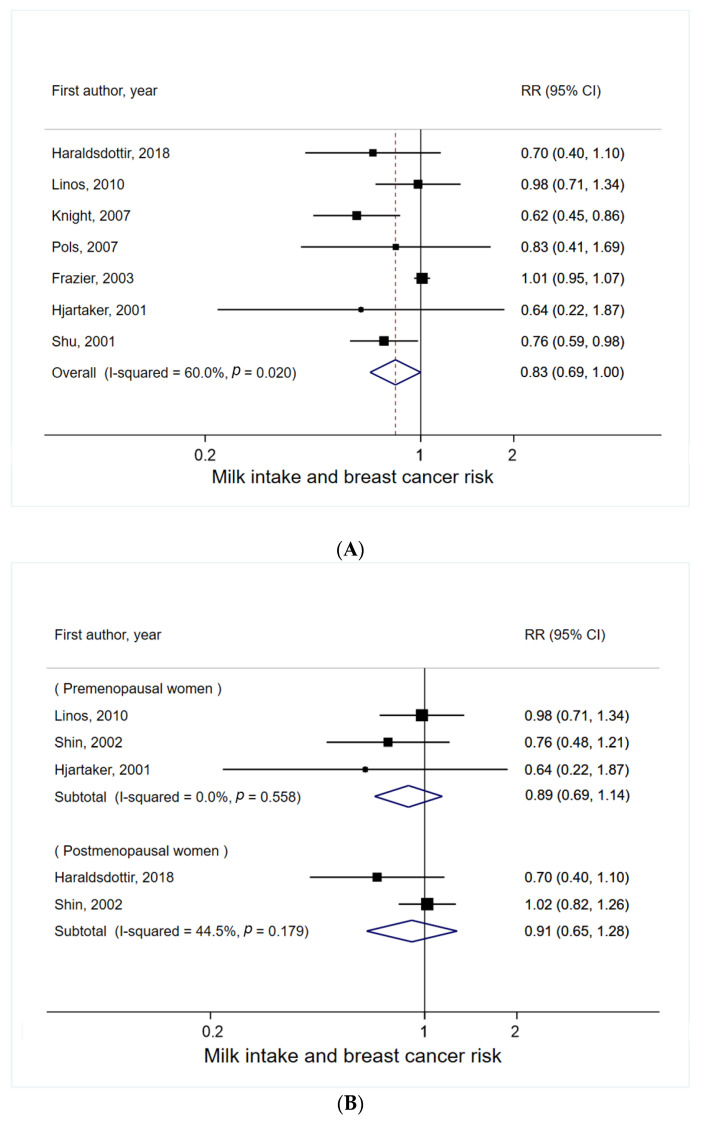
Meta-analysis of milk intake in early life (**A**) with any breast cancer risk, (**B**) by menopausal status, (**C**) by cancer stage, (**D**) by fat content of milk, (**E**) by life stage of milk intake, and (**F**) by study design. For each study, the RR and corresponding 95% CI of individual studies are marked by the location of the black box and the width of the horizontal line, respectively. The size of the box indicates the weight that the RR receives when calculating the summary estimate. The summary RR and its 95% CI are represented by the center and weight of the diamond, respectively.

**Figure 3 nutrients-14-01233-f003:**
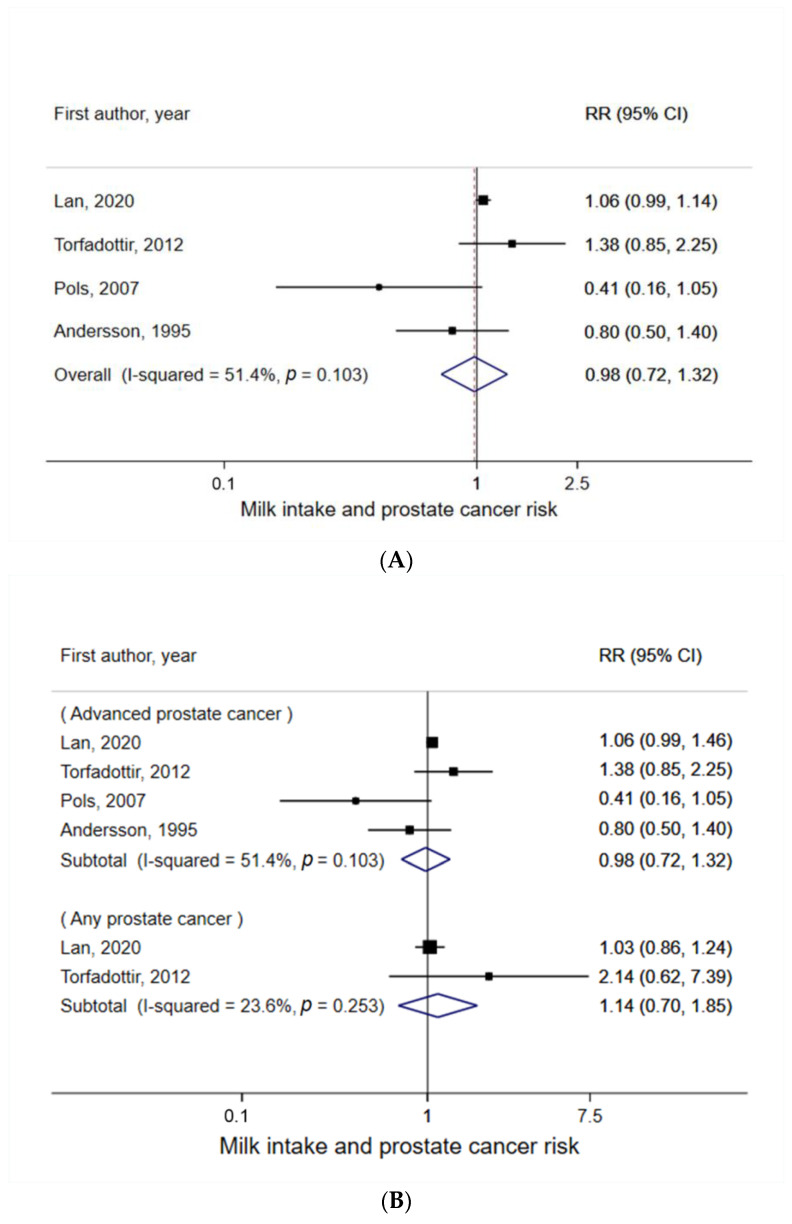
Meta-analysis of milk intake in early life (**A**) with any prostate cancer risk, (**B**) by cancer stage, (**C**) by life stage of milk intake, and (**D**) by study design. For each study, the RR and corresponding 95% CI of individual studies are marked by the location of the black box and the width of the horizontal line, respectively. The size of the box indicates the weight that the RR receives when calculating the summary estimate. The summary RR and its 95% CI are represented by the center and weight of the diamond, respectively. * I^2^ and *p* value are not defined when there is only one study in the meta-analysis.

**Figure 4 nutrients-14-01233-f004:**
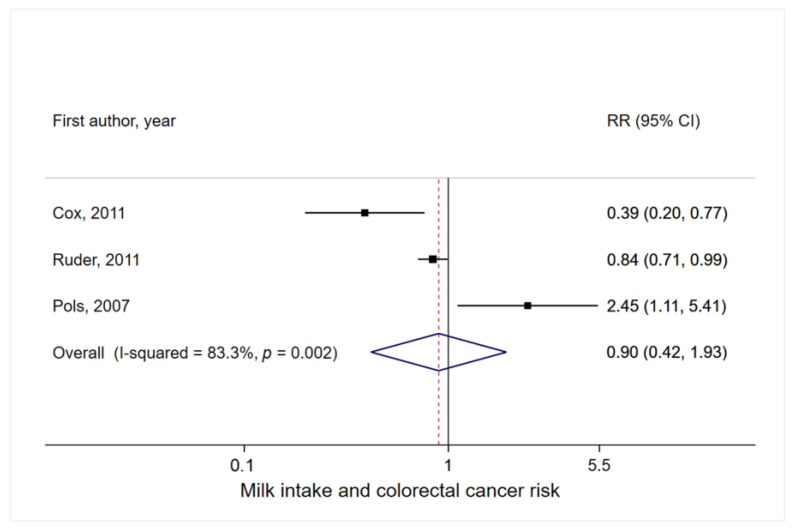
Meta-analysis of milk intake in early life with any colorectal cancer risk. For each study, the RR and corresponding 95% CI of individual studies are marked by the location of the black box and the width of the horizontal line, respectively. The size of the box indicates the weight that the RR receives when calculating the summary estimate. The summary RR and its 95% CI are represented by the center and weight of the diamond, respectively.

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
