# Peer review of "Milk Intake in Early Life and Later Cancer Risk: A Meta-Analysis"

_nutrients, 2022, doi:10.3390/nu14061233_

Round 1

Reviewer 1 Report

Line 69 – Why were review articles excluded?

Figure 1 – Please justify/explain better the exclusion of “ irrelevant exposure or outcome”.

Line 81 – Were is written “important characteristics of the study population (e.g., first author…). What is the importance of the name of the first author to characterize the studied population?

Line 84 – This is not clear: “For the study selection and data extraction, two authors (HG and QYC) independently participated, and any disagreement was resolved through discussion with NK”. The meaning of HG, QYC and NK is also not clear.

Line 89 – Recently there is a tendency in many papers to simply mention statistical methods. In this case: “… DerSimonian-Laird random effects model. Potential for small study effects, such as publication bias, was assessed using Egger's test”…and so on. Just as a suggestion, I think that a brief explanation of the principles, application and output interpretation of these methodologies would turn the paper friendly  for more readers. Some of them might not be familiar with these statistical tools in particular. The same rational applies to the legend of Figure 2. It should be kept in mind that despite statistics are crucial tools to explore results, the core of the theme is nutrition and health effects, and this is what most matters to the readers. Explaining briefly the statistical methos is also a way to share knowledge with more researchers in similar areas. 

Line 197 -  The expression “publication bias” should be explained.

Line 211 – Where is written “Furthermore, with dairy intake in adulthood, a meta-analysis observed that a significant inverse association with breast cancer risk was pronounced in Asian populations compared to Western populations”, formulating a hypothesis to explain this outcome would enrich the article.

Many compounds present in milk are known to modulate cancer processes. Discussing calcium is a very relevant issue, but its choice should be justified. Additionally, considering the role of microRNAs in cancer and that they are highly enriched in cow milk, their inclusion in the discussion could be quite interesting.

Reviewer 2 Report

The authors have proposed an interesting review on milk and derivatives. The study is well conducted. The results presented are of interest, in fact the frequentely consumption of these foods in adults is associated with breast and prostate cancer. It is extremely interesting that the same consumption in childhood and adolescence is not associated with risk.
In my opinion, another randomized studues are needed to confirm the results. 
